# Spectrum of Genes for Non-*GJB2*-Related Non-Syndromic Hearing Loss in the Russian Population Revealed by a Targeted Deafness Gene Panel

**DOI:** 10.3390/ijms232415748

**Published:** 2022-12-12

**Authors:** Olga Shatokhina, Nailya Galeeva, Anna Stepanova, Tatiana Markova, Maria Lalayants, Natalia Alekseeva, George Tavarkiladze, Tatiana Markova, Liudmila Bessonova, Marina Petukhova, Daria Guseva, Inga Anisimova, Alexander Polyakov, Oxana Ryzhkova, Elena Bliznetz

**Affiliations:** 1Federal State Budgetary Institution “Research Centre For Medical Genetics”, 115478 Moscow, Russia; 2Federal State Budgetary Institution of Science “National Research Centre for Audiology and Hearing Rehabilitation”, 117513 Moscow, Russia; 3Federal State Budgetary Educational Institution of Further Professional Education “Russian Medical Academy of Continuous Professional Education”, 125993 Moscow, Russia

**Keywords:** hearing loss, non-syndromic, non-*GJB2* related, MPS, panel sequencing

## Abstract

Hearing loss is one of the most genetically heterogeneous disorders known. Over 120 genes are reportedly associated with non-syndromic hearing loss (NSHL). To date, in Russia, there have been relatively few studies that apply massive parallel sequencing (MPS) methods to elucidate the genetic factors underlying non-*GJB2*-related hearing loss cases. The current study is intended to provide an understanding of the mutation spectrum in non-*GJB2*-related hearing loss in a cohort of Russian sensorineural NSHL patients and establish the best diagnostic algorithm. Genetic testing using an MPS panel, which included 33 NSHL and syndromic hearing loss (SHL) genes that might be misdiagnosed as NSHL genes, was completed on 226 sequentially accrued and unrelated patients. As a result, the molecular basis of deafness was found in 21% of the non-*GJB2* NSHL cases. The total contribution pathogenic, and likely pathogenic, variants in the genes studied among all hereditary NSHL Russian patients was 12%. *STRC* pathogenic and likely pathogenic, variants accounted for 30% of diagnoses in *GJB2*-negative patients, providing the most common diagnosis. The majority of causative mutations in *STRC* involved large copy number variants (CNVs) (80%). Among the point mutations, the most common were c.11864G>A (p.Trp3955*) in the *USH2A* gene, c.2171_2174delTTTG (p.Val724Glyfs*6) in the *STRC* gene, and c.107A>C (p.His36Pro) and c.1001G>T (p.Gly334Val) in the *SLC26A4* gene. Pathogenic variants in genes involved in SHL accounted for almost half of the cases with an established molecular genetic diagnosis, which were 10% of the total cohort of patients with non-*GJB2*-related hearing loss.

## 1. Introduction

Hearing loss is one of the most common congenital abnormalities. On average, 1 in 500 infants worldwide is diagnosed with severe bilateral sensorineural hearing loss [1,2]. According to data from the audiological screening, childhood hearing loss affects 3 per 1000 newborns in Russia [3].

At least 60% of childhood sensorineural hearing loss is caused by genetic factors, of which 80% is NSHL with isolated hearing impairment, and 20% is SHL with other abnormalities [4,5]. Pathogenic variants in the DFNB1 locus containing the *GJB2* gene, encoding the connexin 26 protein (MIM#220290), are the most commonly identified cause of congenital sensorineural NSHL [6]. The proportion of DFNB1 among NSHL cases varies in different populations from 2% to 68% [7]. The proportion of deafness, including DFNB1, caused by mutations in the *GJB2* gene among patients with NSHL in Russia is 46%. The frequency of *GJB2*-related deafness in Russia is 1:1000 [8].

Hearing loss is an extremely heterogeneous disease. Currently, over 120 genes are reportedly associated with NSHL (Hereditary Hearing Loss Homepage: https://hereditaryhearingloss.org/ (accessed on 31 October 2022)), and over 400 are associated with SHL [9]. In a multi-ethnic sample of patients from Europe, America, and Asia, studied using MPS, the proportion of non-*GJB2*-related hearing loss was 31% [10]. In a sample of patients from France, this was 24% [11].

Syndromal genes of hearing loss are characterized by a wide variety of clinical features. The most common genes of SHL are often clinically difficult to distinguish from NSHL due to the syndromic features developing later than hearing loss. Some genes are characterized by features that require special diagnostic tests to detect [10]. In addition, mutations in one gene can cause both SHL and NSHL. The genes of SHL that are identified in patients with a referring diagnosis of NSHL, according to worldwide data, are Usher syndrome, Pendred syndrome, Deafness-infertility syndrome, Alström syndrome, autosomal dominant non-ocular Stickler syndrome, branchio-oto-renal syndrome, MYH9-associated disease, and Wolfram syndrome [10].

In recent years, MPS, including targeted panels, whole/clinical exome sequencing, and whole genome sequencing, has revolutionized the genetic screening of disorders with high genetic heterogeneity, such as hearing impairment. This method allows for the effective screening of many genes in one test [12]. However, to date, in Russia, there are relatively few studies that focus on applying MPS methods to elucidate the genetic factors underlying non-*GJB2*-related hearing loss. Thus, the prevalence and spectrum of other genes of hearing loss among Russian patients remain unknown [13].

In order to incorporate MPS into clinical practice, there is an urgent need to select the most efficient, inexpensive, and quickest method. In this study, we assessed the effectiveness of a custom panel for sensorineural NSHL in Russia. We enrolled 226 patients who were clinically diagnosed with sensorineural NSHL. Part of this cohort was previously described in three studies conducted by Markova et al. and Lalayants et al. that presented the phenotypes and the audiological features of hearing loss caused by mutations in the *STRC*, *USH2A*, and *OTOF* genes [14,15,16]. The current study is intended to provide an understanding of the mutation spectrum in non-*GJB2*-related hearing loss in a cohort of Russian sensorineural NSHL patients and establish the best diagnostic algorithm.

## 2. Results

### 2.1. Genetic Diagnoses

In the first step, DNA samples from 226 probands with NSHL were tested using the MPS panel. As a result, pathogenic and likely pathogenic point variants were identified in 31 patients. All variants were classified according to the ACMG criteria [17,18] and submitted to ClinVar (Submission numbers: SCV002754430-SCV002754457 and SCV002756424-SCV002756450).

Our MPS panel includes the *STRC* and *OTOA* genes. According to global data, the majority of mutant alleles (50–77% in different ethnic groups) being gross deletions [19]. Since the AmpliSeq data are not suitable for the detection of CNVs [20], in the second step, we focused on a targeted study of the *STRC* and *OTOA* genes to detect overlapping zones of biallelic gross deletions using the IGV software. As a result, eight patients were found to lack unambiguously aligned reads of the genomic sequence in which the *STRC* gene was localized (Figure 1). Moreover, because the *STRC* and *OTOA* genes have pseudogenes, we looked for mutations in non-uniquely mapped reads that were filtered out during data processing but were also visualized by the IGV software. As a result, the c.4057C>T mutation (p.Gln1353*) in the homozygous state was found in one patient in exon 25 of the *STRC* gene in the region of ambiguously mapped reads (Figure 2). Normally, these reads are uniquely aligned due to the pseudogene having several SNPs, including an identified mutation. In order to determine the localization of this variant, we selected primers specific to this gene. The primers were synthesized taking into account the differences between the *STRC* and *pSTRC* genes in three nucleotides, intron 19, and intron 21. As a result of direct automatic Sanger sequencing, the c.4057C>T mutation (p.Gln1353*) was detected in the homozygous/hemizygous state and was shown to be localized in the *STRC* gene.

In the third stage, CNVs detection using the MLPA method was performed for several groups of patients. First, we confirmed the presence of biallelic *STRC* gene deletions in eight patients from the second stage (Figure 1). We also tested six patients with homozygous/hemizygous point variants in the *STRC* and *OTOA* genes to analyze their zygosity. As a result, gross deletions at the second allele were detected in all patients. Since CNVs account for 6–9% and 10% of all pathogenic and likely pathogenic variants in the *USH2A* and *SLC26A4* genes, respectively [21,22], we also tested 29 patients that only had one variant (pathogenic, likely pathogenic, or a variant of uncertain significance) in the *USH2A* genes and 9 patients with one variant in the *SLC26A4* genes. As a result, gross deletions of one or more exons in the *USH2A* gene were found in three patients.

For variants of uncertain significance, a family analysis was performed to confirm the variants’ segregation with the HL among the family members. Some families were not analyzed due to the lack of biological material of relatives. Trans-position for six variants was shown by examining the probands’ parents in seven families.

### 2.2. Analysis of Causative Variants

Thus, after all steps, among 226 patients with non-GJB2 NSHL, a genetic cause of the disease was found in 48 cases (21%). A total of 55 pathogenic/likely pathogenic variants were identified (see Appendix A).

A total of 21 of the 55 identified variants (38%) had not been previously described. Eight variants were identified in more than one unrelated family. Two variants occurred in the homozygous state. Among the repeated variants, there were five that had been previously described (c.2171_2174del, g.(_43890333)_(43940887_?)del in the *STRC* gene [23,24], c.11864G>A and c.12234_12235del in the *USH2A* gene [25,26], c.1001G>T in the *SLC26A4* gene [27,28] and three that had not been previously described (c.107A>C in the *SLC26A4* gene, c.2656del and c.4903A>T in the *OTOF* gene). Families carrying these variants were from different regions of the Russian Federation.

The most common pathogenic variant of the *USH2A* gene was the c.11864G>A mutation (p.Trp3955*), which accounted for half of the mutant alleles (9/18). In an earlier study of another sample of Russian patients with Usher syndrome, the p.Trp3955* mutation was found in 30% (6/28) of mutant gene alleles, which was not statistically different from the results of this study (ꭓ^2^ = 2.182, *p* = 0.14) [29].

An accumulation of the c.2171_2174delTTTG variant (p.Val724Glyfs*6) in exon 5 of the *STRC* gene was observed in 1.8% (4/226) of our cohort of Russian patients. This mutation was found in 10% (4/28) of pathogenic gene alleles.

Among mutations in the *SLC26A4* gene, the previously undescribed c.107A>C (p.Glu29Gln) mutation was found to be the most frequent, accounting for 27% (3/11) of mutant alleles of the pendrin gene. The c.1001G>T splice-site mutation, which is found at 18% of pathogenic alleles of the pendrin gene (2/11), was shown to be the second most frequent variant in the *SLC26A4* gene.

In the present study, CNVs were the cause of 38% of diagnoses. In this case, 8 patients had CNVs detected at both alleles and 10 patients had CNVs in compound heterozygosity with a second pathogenic variant. The proportion of CNVs was significantly higher than that reported globally (ꭓ^2^ = 5.013, *p* = 0.026). Gross deletions were observed in 80% of the mutant alleles of the *STRC* gene, 50% of the *OTOA* gene, and 17% of the *USH2A* gene.

The novel variants identified in this study were categorized according to the guidelines of ACMG (Table 1) and submitted to ClinVar.

### 2.3. Genetic Heterogeneity of NSHL

In this study, 13 genes of hereditary hearing loss were identified among the patients with NSHL. Pathogenic and likely pathogenic variants were found in 12 genes of autosomal recessive hearing loss, representing 90% of confirmed cases (43/48), in 2 genes of autosomal dominant hearing loss (5% (3/48)), and in 1 gene of X-linked recessive form (5% (2/48)).

The contribution of the genes studied in this work to non-*GJB2*-related hearing loss was 21% (48/226) (Figure 3). The calculation of the total contribution the studied genes among all hereditary NSHL is presented in the Discussion section.

The total number of genes detected in this study is 13. The proportions of each of the genes of hearing loss do not statistically differ from the world data (Table 2). It emerged that more than 70% of all diagnoses were attributable to five genes: *STRC* (29%), *USH2A* (17%), *SLC26A4* (12%), *MYO7A* (10%), and *OTOF* (8%). In the other eight genes, mutations were observed in 1–2 cases. The *STRC* (ꭓ^2^ = 10.800, *p* = 0.002) and USH2A (ꭓ^2^ = 4.019, *p* = 0.046) genes had significantly higher contributions as compared to the *TECTA, POU3F4, TMPRSS3, ADGRV1, MYO15A, OTOA, PTPRQ*, and *TMC1* genes. According to data reported in the literature, mutations in the *STRC* gene contribute significantly more than the other genes identified in this work. The prevalence of the observed deafness genes above among the Russian population is similar to the prevalence reported in other populations. Thus, mutations in the *STRC* gene were more frequently found in *GJB2*-negative patients, but the level did not reach statistical significance. 

## 3. Discussion

Hearing loss is one of the most genetically heterogeneous disorders known [9]. In total, 60% of cases are believed to be of genetic origin, and 30% of these are thought to be syndromic. Little is known about the frequency of NSHL variants in Russians, except for the preeminent relevance of *GJB2* mutations. The current study was the first to analyze the frequency of various non-*GJB2*-related hearing loss genes in a large cohort of Russian patients.

Genetic testing using an MPS panel, which included 33 NSHL and SHL genes misdiagnosed as NSHL genes, was completed on 226 sequentially accrued and unrelated patients with sensorineural NSHL. Primary analysis of panel sequencing data revealed the cause of the disease in 29 of 226 patients (13%). Moreover, in a targeted study, we also focused on the *STRC* and *OTOA* genes because these genes are common sites for large deletions and there are highly homologous pseudogenes for these genes. Targeted analysis of these genes for biallelic gross deletions and mutations in homologous regions increased this rate. The cause of the disease was identified in nine additional patients, increasing the diagnostic value to 17% (38/226). The search for CNVs in patients with one heterozygous variant in the *USH2A*, *SLC26A4*, *STRC*, and *OTOA* genes increased the diagnostic rate to 18% (41/226). A familial analysis established the pathogenicity of five variants, allowing the cause of hearing loss to be determined in an additional seven patients. Therefore, the overall diagnostic rate was 21% (48/226). The results showed the crucial importance of studying CNVs and mutations in highly homologous regions of genes. The use of these steps increased the diagnostic value by a factor of 1.5 as compared with the primary analysis. In familial cases, the diagnosis was confirmed for 19% of cases (9/48). In isolated cases of hearing loss in the family, the diagnostic rate was 22% (39/178). Thus, no difference was found between diagnostic efficacy in sporadic cases of hearing loss compared to familial cases (ꭓ2 = 0.0226, *p* = 0.635).

In addition, CNVs detection in the *USH2A* and *SLC26A4* genes using the MLPA method was performed since this type of mutation is common for these genes. After all the steps, a family analysis was performed to explore the pathogenicity of certain variants. As a result, the molecular basis of deafness was found in 21% (48/226) of cases with NSHL. No differences were found between the diagnostic rates of molecular genetic testing in sporadic cases of hearing loss compared to familial cases. The results showed the crucial importance of studying CNVs and mutations in highly homologous regions of genes. The use of these steps increased the diagnostic value by a factor of 1.5 as compared with the primary analysis.

Our results contribute to defining the mutation spectrum in the Russian population, underlining the NSHL genetic heterogeneity. A total of 55 causative variants in 48 patients affected 13 different genes. The total contribution of the genes studied in this work to non-*GJB2*-related hearing loss was 21% (48/226). Previously, we investigated the mutation spectrum and the frequency of *GJB2* hearing loss among 2569 Russian patients with NSHL [30]. According to this study of Bliznetz et al., the frequency of *GJB2* mutations is 43% (1104/2569), and the frequency of non-*GJB2*-related hearing loss is 57% (1465/2569). Our cohort of Russian patients investigated using the MPS panel is part of the one described in the study of Bliznetz et al. In this case, 226 patients were randomly selected for the present work from 1465 patients without pathogenic and likely pathogenic variants in the *GJB2* gene. As a result, causative variants were identified in 21% of cases (48/226). Thus, the prevalence of the genes studied in this work among all hereditary NSHL in Russia is equal to 12%. This prevalence was calculated by taking 21% of the 57% (1465 of 2569 cases without *GJB2* mutation). Among the 1119 NSHL patients from Europe, America, and Asia, and 207 NSHL patients from France mutations in the 33 genes studied in our work were more common than in the Russian sample: 29% (ꭓ^2^ = 21.191, *p* < 0.001) [10], and, 23% (ꭓ^2^ = 12.120, *p* < 0.001), respectively [11]. However, because of the higher frequency of *GJB2*-related hearing loss in the Russian Federation, the overall prevalence of mutations in the *GJB2* gene and the 33 studied genes among the NSHL Russian patients was higher (55%) than in these cohorts (Figure 3). In a multi-ethnic sample of 1119 patients from Europe, America, and Asia, this was 37% (416/1119, ꭓ^2^ = 37.982, *p* < 0.001) [10], and, in a sample of 207 patients from France, this was 46% (96/207) ꭓ^2^ = 3.923, *p* = 0.048) [11]. The estimated frequency of DFNB1 in the Russian Federation is 1:1000 [8]. Accordingly, in this study the prevalence of non-*GJB2*-related mutations would be 1 per 3500 newborns.

The prevalence of each gene among all detected genes in Russian patients with NSHL did not differ from the global data. *STRC* pathogenic and likely pathogenic variants accounted for 30% (14/48) of diagnoses in *GJB2*-negative patients, providing the most common diagnosis. Mutations in the *STRC*, *USH2A*, *SLC26A4*, *MYO7A*, and *OTOF* genes accounted for 1 to 3% of all NSHL in Russian patients. Mutations in each of the other genes such as *TECTA MYO15A*, *POU3F4*, *TMPRSS3*, *PTPRQ*, *ADGRV1*, *TMC1*, *ACTG1*, and *OTOA* accounted for less than 1%.

There were peculiarities in the mutation spectrum among Russian patients in three genes, those being *STRC*, *USH2A*, and *SLC26A4*. The pathogenic variant c.11864G>A (p.Trp3955*) was observed in 50% of mutant alleles of the *USH2A* gene and the deletion c.2171_2174delTTTG (p.Val724Glyfs*6) was observed in 10% of mutant alleles of the *STRC* gene. Two repeated mutations c.107A>C and c.1001G>T, were 25% and 15% of mutant alleles of the *SLC26A4* gene, respectively.

It is noteworthy that the majority of causative mutations in *STRC* involved large CNVs (80%), underscoring the requirement that all comprehensive genetic testing panels for hearing loss include CNVs detection. In addition, gross deletions accounted for 50% of the *OTOA* gene, and 17% of the *USH2A* gene. The proportion of alleles with CNVs found in Russian patients with NSHL was significantly higher than that reported in world data, which may indicate a higher frequency of this type of mutation among Russian patients or the omission of point mutations, probably due to the presence of uncovered gene coding regions and the limitations of the IonS5 platform, i.e., the difficulty of sequencing homopolymer regions.

In the study sample of patients with NSHL, 50% (23/48) received a genetic diagnosis implicating a misdiagnosed SHL gene. Thus, syndromic genes accounted for 10% of the total sample of patients with non-*GJB2*-related hearing loss. Most of the syndromic diagnoses masquerading as NSHL were related to Usher syndrome (eight patients with causative mutations in the *USH2A* gene, three in the *MYO7A* gene, and one in the *ADGRV1* gene). In the sample of patients with NSHL, syndromes such as Pendred syndrome (six in the *SLC26A4* gene), deafness, and male infertility syndrome (DIS) (three males and two females with NSHL with bi-allelic gross deletions in the *STRC* and *CATSPER2* genes) were also found. The genetic diagnosis of NSHL mimics is crucially valuable. It provides prognostic information on the possible progression of hearing loss, permits meaningful genetic counseling, and impacts treatment decisions [10,31]. The clinical features of hearing loss caused by *STRC* and *USH2A* gene mutations that are most frequent in the Russian population were presented in our previous papers [14,15,16].

The genetic diagnosis of hereditary hearing loss is highly difficult due to its enormous underlying genetic heterogeneity. The implementation of our MPS panel containing 33 genes improved the management of our patients, as it allowed us to detect causative variants in 13 different genes. However, pathogenic variants in a few genes still explain a great number of hearing loss cases. The main example is *GJB2*, which encodes connexin 26. A system for searching for six frequent mutations in the *GJB2* gene at the RCMG allowed for the detection of two mutations in 41% of patients with senso-neural NSHL and at least one mutation in 50% of patients [8]. Our study of the mutation spectrum of other genes among Russian patients showed that gross deletions/duplications in the *STRC* and *USH2A* genes and the four most frequent mutations (c.11864G>A in the *USH2A* gene, c.2171_2174delTTTG in the *STRC* gene and c.107A>C and c.1001G>T in the *SLC26A4* gene) accounted for 38% of diagnoses framed with the developed MPS panel.

Perhaps the best option for the first stage is to detect high frequency mutations of *GJB2*, *STRC*, *USH2A*, and *SLC26A4* genes (including CNVs), which is probably to detect two mutations in a significant proportion of patients with NSHL using MLPA methods. This will significantly reduce the cost of diagnosis and shorten the waiting time for diagnosis for many patients. In the next step, the optimal option is whole exome sequencing, which is currently no more expensive than panel sequencing. In addition, this method makes it possible to examine all currently known hearing loss genes, which theoretically increases diagnostic efficiency. Of course, whole exome sequencing is currently the preferred method for diagnosing hereditary hearing loss for all population because it allows identify variants in all known hearing loss genes. However, the country-specific algorithms can reduce the cost of diagnosis and speed up diagnosis in many cases.

## 4. Materials and Methods

### 4.1. Clinical Data

The current study analyzed the DNA of 226 unrelated patients with sensorineural NSHL, aged 6 months to 50 years, with varying severity of hearing loss, who had no mutations in the *GJB2* gene as a result of routine diagnostic testing according to the protocol described earlier [8]. Of the 226 cases, 48 were familial, 178 isolated. Of 226 patients, 5 were referred with a diagnosis of bilateral sensorineural NSHL and an enlarged vestibular aqueduct (EVA, as revealed by temporal bone computed tomography), and 18 patients with NSHL and auditory neuropathy. The study also included 4 affected and 12 unaffected relatives of the probands. Most patients were examined at the Research and Counseling Department of the Research Centre for Medical Genetics (RCMG) and at the National Research Centre for Audiology and Hearing Rehabilitation. Patients or patient’s parents provided oral and written consent for this study, approved by the ethics committee of RCMG.

### 4.2. Genetic Testing

Genomic DNA was extracted from whole venous blood via a Wizard^®^ Genomic DNA Purification Kit (Promega, Madison, WI, USA) following the manufacturer’s protocol.

The DNA of patients was studied using the MPS method via an Ion S5 next-generation sequencer (Thermo Fisher Scientific, Waltham, MA, USA) with an Ion AmpliSeq™ Library Kit 2.0 following the manufacturer’s protocol. Samples were prepared using ultra rapid multiplex PCR technology combined with subsequent sequencing (AmpliSeq™). The genes that were included in this panel were selected according to the prevalence reported and the results of exome sequencing of Russian patients [10,32,33,34,35]. In this case, 29 genes included in the panel are the most common causes of NSHL according to world data. We also added four SHL genes in which mutations have been identified in patients with a clinical diagnosis of NSHL. Of the 33 genes, 21 genes are the cause of NSHL only (*MYO15A*, *TECTA*, *TMPRSS3*, *TMC1*, *COL11A2*, *OTOF*, *OTOA*, *KCNQ4*, *LOXHD1*, *MYH14*, *MYO6*, *ACTG1*, *PTPRQ*, *OTOGL*, *TRIOBP*, *CLDN14*, *LRTOMT*, *PJVK* (*DFNB59*), *TPRN*, *POU3F4*, *SMPX*), 8 genes are the cause of both NSHL and SHL (*MYO7A*, *CDH23*, *WHRN*, *PCDH15* (Usher s.), *WFS1* (Wolfram s.), *COL11A2* (Stickler s.), *SLC26A4* (Pendred s.)), *MYH9* (MYH9-associated disease), and 4 genes are the cause of SHL (*USH2A*, *ADGRV1* (Usher s.), *EYA1* (Branchiootic s.), *ALMS1* (Alstrom s.)). The genes included in the panel and their NCBI cDNA reference sequences are shown in Table 3.

Sequencing data were processed using a standard computer-based algorithm from Thermo Fisher Scientific (Torrent Suite™) and NGSData software (http://ngs-data.ru/ (accessed on 31 October 2022)). Sequenced fragments were visualized via Integrative Genomics Viewer (IGV) software (© 2013–2018 Broad Institute, and the Regents of the University of California, San Diego, CA, USA). The Genome Aggregation Database (gnomAD v.2.1.1) was used to determine the allele frequency of newly discovered variants.

Analysis of gross deletions and duplications in the *STRC*, *USH2A*, *SLC26A4*, and *OTOA* genes was conducted via the MLPA^®^ method (MRC-Holland, the Netherlands) using SALSA MLPA P461–A1 (*STRC* and *OTOA*), P361–A2, P362–A2 (*USH2A*), and P280–B3 (*SLC26A4*) kits following the manufacturer’s protocol. MLPA data were analyzed with the Coffalayser software (MRC-Holland, v.220513.1739).

The revealed gene modifications were designated in accordance with HGVS nomenclature (https://varnomen.hgvs.org/ (accessed on 31 October 2022)).

The following online prediction programs were utilized to determine pathogenicity in silico: Mutation Taster (http://www.mutationtaster.org/ (accessed on 31 October 2022)), UMD-predictor (http://umd-predictor.eu/ (accessed on 31 October 2022)), SIFT/Provean (http://provean.jcvi.org/index.php (accessed on 31 October 2022)), PolyPhen-2 (http://genetics.bwh.harvard.edu/pph2/index.shtml/ (accessed on 31 October 2022)), and Human Splicing Finder (http://www.umd.be/HSF/ (accessed on 31 October 2022)).

The American College of Medical Genetics and Genomics (ACMG) guidelines for interpretation of massive parallel sequencing data [17,18] were used to define the clinical significance of newly discovered variants. All reported variants were submitted to ClinVar (SCV002754430-SCV002754457, Submission ID: SUB12315120, SUB12323588).

## 5. Conclusions

NSHL is a heterogeneous group of diseases, the etiological diagnosis of which is difficult to establish by traditional clinical examination. Such difficulties are attributed to the high number of described genes (a number that is still growing), the similarity of the clinical picture, and syndromal genes mimicking NSHL. All of the above factors emphasize the need to bring molecular diagnostics of hearing loss up-to-date. Applying modern genomic sequencing technologies in this study allowed us to partially elucidate the genetic landscape of NSHL in Russia. This paper presented the possibilities of molecular genetic diagnosis in Russian patients with hereditary hearing loss and the possible cost-effective algorithm for their investigation.

## Figures and Tables

**Figure 1 ijms-23-15748-f001:**
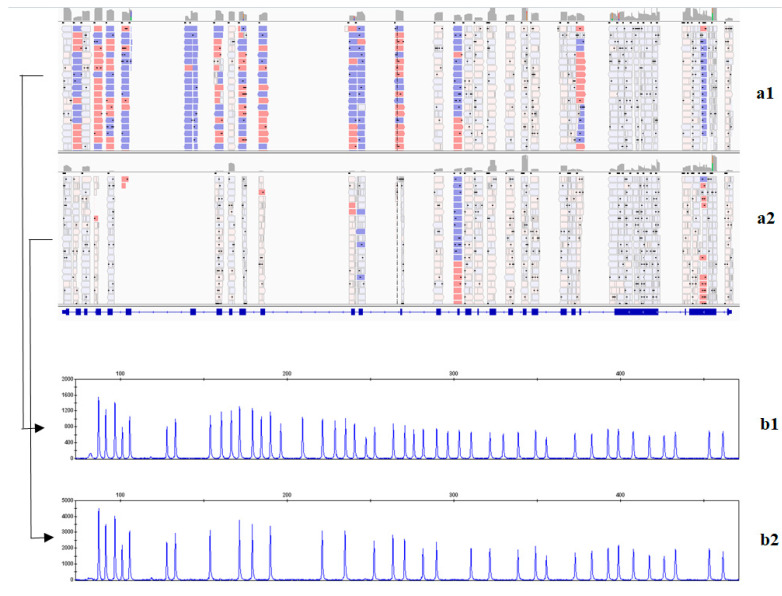
The detection of biallelic gross deletions of the *STRC* gene. (**a**) Using IGV genomic browser, (**b**) using MLPA, (**a1**,**b1**)—normal, (**a2**)—no reads aligned with the *STRC* gene sequence, (**b2**)—*STRC, CKMT1B, CATSPER2* deletions in the homozygous state.

**Figure 2 ijms-23-15748-f002:**
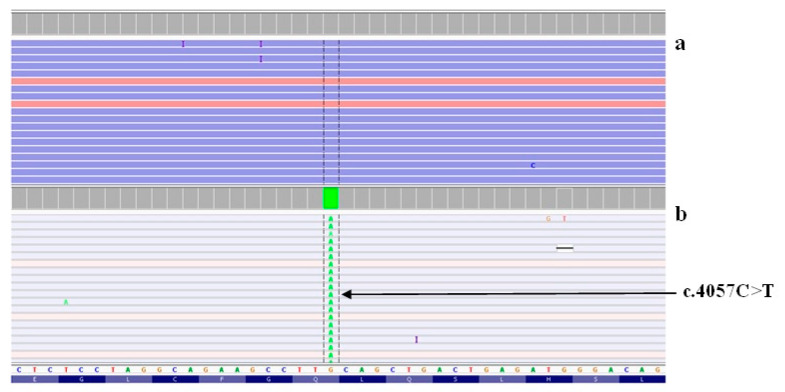
The alignment of reads in the region of exon 20 of the *STRC* gene visualized in the IGV genomic browser. (**a**) Normal, and (**b**) c.4057C>T mutation.

**Figure 3 ijms-23-15748-f003:**
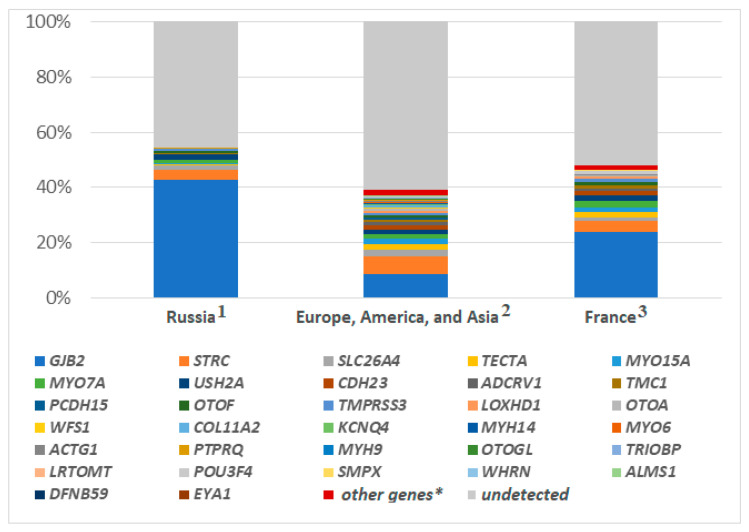
The contribution of the studied genes of the NSHL: 1—among Russian patients; 2—in a multi-ethnic sample of patients from Europe, America, and Asia [10]; 3—in a sample of patients from France [11]; *—genes not studied in this study.

**Table 1 ijms-23-15748-t001:** Pathogenicity of the novel variants: a—designations are given in accordance with the ACMG recommendations for the interpretation and classification of nucleotide sequence variants [17,18]. b—variants identified in the same patients are presented in the articles by Markova et al. and Lalayants et al. [14,15,16].

Gene	Variant	Pathogenicity	Criteria ^a^	The ClinVar Accession Numbers
*ADGRV1*	c.6610C>T (p.Gln2204*)	Likely pathogenic	PVS1, PM2	SCV002754430
*ADGRV1*	c.10198C>T (p.Gln3400*)	Likely pathogenic	PVS1, PM2	SCV002754431
*MYO15A*	c.8910del (p.Val2971fs*63)	Likely pathogenic	PVS1, PM2	SCV002754432
*MYO7A*	c.1738_1745del (p.Val581Leufs*28)	Likely pathogenic	PVS1, PM2	SCV002754433
*MYO7A*	c.3893G>A (p.Gly1298Glu)	Likely pathogenic	PM2, PM3, PM5, PP3	SCV002754434
*MYO7A*	c.3612_3615del (p.Ser1205Argfs*26)	Likely pathogenic	PVS1, PM2	SCV002754435
*MYO7A*	c.4528G>A (p.Glu1510Lys)	Likely pathogenic	PM2, PM3, PP2, PP3	SCV002754436
*OTOA*	c.562_569dup (p.Phe191fs*48)	Likely pathogenic	PVS1, PM2	SCV002754437
*OTOF*	c.2656del (p.Val886Serfs*114) ^b^	Likely pathogenic	PVS1, PM2	SCV002754438
*OTOF*	c.5169_5170del (p.Ile1724Leufs*19) ^b^	Likely pathogenic	PVS1, PM2	SCV002754439
*OTOF*	c.4903A>T (p.Arg1635*) ^b^	Likely pathogenic	PVS1, PM2	SCV002754440
*OTOF*	c.2214+5G>C ^b^	Likely pathogenic	PM2, PM3, PP3, PP4	SCV002754441
*POU3F4*	c.983A>G (p.Asn328Ser)	Likely pathogenic	PS1, PM2, PM5, PP3	SCV002754443
*PTPRQ*	c.1291C>T (p.Arg431*)	Likely pathogenic	PVS1, PM2	SCV002754445
*SLC26A4*	c.107A>C (p.His36Pro)	Likely pathogenic	PM2, PM3, PP3, PP4	SCV002754447
*SLC26A4*	c.208C>T (p.Pro70Ser)	Likely pathogenic	PM2, PM3, PP4, PP5	SCV002754448
*STRC*	g.(?_43906612)_(43906674_)?del (delSTRC, ex5) ^b^	Pathogenic	PVS1, PM2, PM3	SCV002754455
*TECTA*	c.2458A>T (p.Lys820*)	Likely pathogenic	PVS1, PM2	SCV002754449
*TMC1*	c.1750C>T *p.Gln584*)	Likely pathogenic	PVS1, PM2	SCV002754450
*USH2A*	g.(?_216108034)_(216108100_?)del (delUSH2A, ex38) ^b^	Likely pathogenic	PVS1, PM2	SCV002754452
*USH2A*	g. (?_216462679)_(216462739_?)del (delUSH2A, ex11)	Likely pathogenic	PVS1, PM2	SCV002754453

**Table 2 ijms-23-15748-t002:** The number of probands with NSHL with each of the identified genes of hearing loss among patients with an established molecular genetic diagnosis (n = 48). *—The comparison was made between the proportions in our cohort of Russian patients and the multi-ethnic sample of patients from Europe, America, and Asia from the study conducted by Sloan-Heggen et al. [10]. (95% CI)—confidence interval with the 95% confidence level.

Gene	Phenotype	Number of Probands	The Proportion of Probands with Established Genes of Hearing Loss		*p* Value *
			Among the 48 Cases of This Study (95%CI)	In Europe, America and Asia *	
*STRC*	DFNB16	14	29% (17–43%)	30%	0.963
*USH2A*	Usher syndrome	8	17% (7–30%)	8%	0.114
*SLC26A4*	DFNB4, Pendred syndrome	6	12% (3–22%)	12%	0.906
*MYO7A*	DFNB2, Usher syndrome/DFNA11	5 (3/2)	10% (3–22%)	8%	0.878
*OTOF*	DFNB9	4	8% (2–20%)	4%	0.324
*TECTA*	DFNB21/DFNA8/12	2 (1/1)	4% (1–17%)	10%	0.625
*POU3F4*	DFNX2	2	4% (0.5–14%)	1%	0.269
*TMPRSS3*	DFNB8/10	2	4% (0.5–14%)	4%	0.769
*ADGRV1*	Usher syndrome	1	2% (0.05–11%)	5%	0.598
*MYO15A*	DFNB3	1	2% (0.05–11%)	9%	0.190
*OTOA*	DFNB22	1	2% (0.05–11%)	3%	0.986
*PTPRQ*	DFNB84	1	2% (0.05–11%)	2%	0.679
*TMC1*	DFNB7/11	1	2% (0.05–11%)	4%	0.769

**Table 3 ijms-23-15748-t003:** The genes included in the MPS panel and their reference sequences.

Gene	RefSeq	Gene	RefSeq
* ACTG1 *	NM_001614	* OTOF *	NM_194248
* ADGRV1 *	NM_032119	* OTOGL *	NM_173591
* ALMS1 *	NM_015120	* PCDH15 *	NM_033056
* CDH23 *	NM_022124	* POU3F4 *	NM_000307
* CLDN14 *	NM_144492	* PTPRQ *	NM_001145026
* COL11A2 *	NM_080680	* SLC26A4 *	NM_000441
*PJVK*	NM_001042702	* SMPX *	NM_014332
* EYA1 *	NM_000503	* STRC *	NM_153700
* KCNQ4 *	NM_004700	* TECTA *	NM_005422
* LOXHD1 *	NM_144612	* TMC1 *	NM_138691
* LRTOMT *	NM_001145308	* TMPRSS3 *	NM_024022
* MYH14 *	NM_024729	* TPRN *	NM_001128228
* MYH9 *	NM_002473	* TRIOBP *	NM_001039141
* MYO15A *	NM_016239	* USH2A *	NM_206933
* MYO6 *	NM_004999	* WFS1 *	NM_006005
* MYO7A *	NM_000260	* WHRN *	NM_015404
* OTOA *	NM_144672	-	-

## Data Availability

The data presented in this study are available on request from the corresponding author. The data are not publicly available due to ethical privacy.

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
