# Peer review of "Spectrum of Genes for Non-GJB2-Related Non-Syndromic Hearing Loss in the Russian Population Revealed by a Targeted Deafness Gene Panel"

_ijms, 2022, doi:10.3390/ijms232415748_

Round 1
Reviewer 1 Report
This manuscript reports on the study of a cohort of 226 Russian patients with apparent non-syndromic hearing loss, in whom a causative role of GJB2 variants had been excluded. The cohort was investigated through the use of a 33-genes panel, in combination with other techniques.
The design of the study could be considered to be relatively obsolete, as it uses the Ion sequencer and a panel with a limited number of genes, whose selection is inevitably subjective. However, the results are interesting, as they provide a glimpse at the genetic epidemiology of NSHL in Russia beyond GJB2, and so they merit publication.
Major comments
11) Lines 181-183. The sentence is not clear. Does ‘family analysis’ mean an analysis of segregation of the variants in parents and other relatives? Was the location in trans confirmed in all supposed-to-be compound heterozygous subjects? This is a critical issue when missense variants are involved.
22) Lines 221-290. Sections 3.3 to 3.5 report results of the study, but merged with issues that should be considered in the Discussion. Please consider rewriting, and avoid repetition.
33) Lines 333-338. Some pathogenic variants were found recurrently in the cohort. Did the families share the same geographic or ethnic origin? Could they share a common-founder? If possible, haplotype analysis for close genetic markers could answer this question.
44) Line 376. ‘Of course, the optimal diagnostic algorithm varies among patients in different countries’. I do not entirely agree with this statement. Currently, the algorithms may be conditioned by cost limitations, which can be country-specific. But the new massive sequencing techniques are able to detect/identify all variants, irrespectively of whether they are more or less population-specific. The sentence should be toned down.
Minor points
11) The acronym for non-syndromic hearing loss contains a recurrent typo (SNHL instead of NSHL) (lines 27, 185, 295 (twice), 307, 313).
22) Lines 33-35. Syndromic forms… Consider rephrasing, it is not clear.
33) Lines 81-82. Spectrum of mutations. ‘of’ is missing.
44) Table 1. DFNB59 is currently known as PJVK.
55) Line 177. ‘CNVs account for 6-9% and 10%...’ of what?
66) Line 178. ’41 patients that had only one variant in these genes’. How many in USH2A, and how many in SLC26A4?
77) Line 206. ‘splicing site mutation’ should read ‘splice-site mutation’.
88) Line 207. ‘in every fifth pathogenic allele? Please rephrase it, I do not understand.
99) Genes should be presented in alphabetical order in Tables 2 and S1.
110) Table 3, 4th row. MYO7A corresponds to DFNB2 (not B3).
111) Line 360. GJB2 should be in italics.
112) Table S1, page 13, fourth row. What does ‘HC’ mean?
113) Table S1, page 13, 14th row. ‘pR1635*’. The three-letter code should be used (Arg).
Author Response
Dear reviewer! Thank you very much for the thorough analysis of our article. I think all the corrections will benefit the article! All changes are highlighted in yellow. The corrections requested by another reviewer are highlighted in blue. Corrections to similar comments are highlighted in green.
Major comments
11) Lines 181-183. The sentence is not clear. Does ‘family analysis’ mean an analysis of segregation of the variants in parents and other relatives? Was the location in trans confirmed in all supposed-to-be compound heterozygous subjects? This is a critical issue when missense variants are involved.
Answer: We performed a familial analysis for variants of uncertain clinical significance, for those families that were available to us. Those missense variants that we could not test due to lack of material were not included. We rewrote this paragraph (Lines 189-1913).
22) Lines 221-290. Sections 3.3 to 3.5 report results of the study, but merged with issues that should be considered in the Discussion. Please consider rewriting, and avoid repetition.
Answer: We moved most of section 3.3 and all of section 3.4 and 3.5 to the discussion section, rewriting them to remove repetition (Lines 262-280, 292-313, 341-352).
33) Lines 333-338. Some pathogenic variants were found recurrently in the cohort. Did the families share the same geographic or ethnic origin? Could they share a common-founder? If possible, haplotype analysis for close genetic markers could answer this question.
Answer: Haplotype analysis was not performed because of the small number of cases in which parental material was available and because the families were from different regions (Lines 204-205).
44) Line 376. ‘Of course, the optimal diagnostic algorithm varies among patients in different countries’. I do not entirely agree with this statement. Currently, the algorithms may be conditioned by cost limitations, which can be country-specific. But the new massive sequencing techniques are able to detect/identify all variants, irrespectively of whether they are more or less population-specific. The sentence should be toned down.
Answer: We rewrote that part (Lines 373-376): «Of course, whole exome sequencing is currently the preferred method for diagnosing hereditary hearing loss for all population because it allows identify variants in all known hearing loss genes. However, the country-specific algorithms can reduce the cost of diagnosis and speed up diagnosis in many cases».
Minor points
11) The acronym for non-syndromic hearing loss contains a recurrent typo (SNHL instead of NSHL) (lines 27, 185, 295 (twice), 307, 313).
Answer: We corrected this throughout the article.
22) Lines 33-35. Syndromic forms… Consider rephrasing, it is not clear.
Answer: We replaced it with «Syndromic genes» (Lines 36-38).
33) Lines 81-82. Spectrum of mutations. ‘of’ is missing.
Answer: We replaced it with «Mutation spectrum» (Line 83). Will that work?
44) Table 1. DFNB59 is currently known as PJVK.
Answer: We corrected it.
55) Line 177. ‘CNVs account for 6-9% and 10%...’ of what?
Answer: We replaced it with (Lines 184-185) «Since CNVs account for 6–9% and 10% of all pathogenic and likely pathogenic variants in the USH2A and SLC26A4 genes, respectively».
66) Line 178. ’41 patients that had only one variant in these genes’. How many in USH2A, and how many in SLC26A4?
Answer: We added that information. There was a typo in the total number, we corrected it. Thank you very much for noticing (Lines 185-187): «we also tested 29 patients that only had one variant (pathogenic, likely pathogenic, or a variant of uncertain significance) in the USH2A genes and 9 patients with one variant in the SLC26A4 genes».
77) Line 206. ‘splicing site mutation’ should read ‘splice-site mutation.
Answer: We corrected it (Line 216).
88) Line 207. ‘in every fifth pathogenic allele? Please rephrase it, I do not understand.
Answer: We corrected it (Lines 216-217): «which is found at 18% of pathogenic alleles of the pendrin gene».
99) Genes should be presented in alphabetical order in Tables 2 and S1.
Answer: We corrected it.
110) Table 3, 4th row. MYO7A corresponds to DFNB2 (not B3).
Answer: We fixed it.
111) Line 360. GJB2 should be in italics.
Answer: We fixed it throughout the article.
112) Table S1, page 13, fourth row. What does ‘HC’ mean?
Answer: We changed it to "SN NSHL" (sensorineural NSHL).
113) Table S1, page 13, 14th row. ‘pR1635*’. The three-letter code should be used (Arg).
Answer: We fixed it.

Reviewer 2 Report
The manuscript named “Comprehensive Genetic Testing of Russian Patients with Non-GJB2 Related Non-Syndromic Hearing Loss” describes a study that is intended to provide an understanding of the mutation spectrum of non-GJB2-related hearing loss in a cohort of Russian sensorineural NSHL patients and to establish the best diagnostic algorithm for this population.
This study emphasizes the value of NGS, including panels, for genetic screening, of disorders with high genetic heterogeneity, such as hearing impairment. As to date, in Russia, there are relatively few studies focusing on NGS methods to elucidate genetic causes underlying hearing loss, and the researchers suggest that NGS is the most efficient, inexpensive, and quickest method for solving deafness in general, and in the Russian population, in particular, and that this method should be incorporated into clinical practice in Russia.
These findings are valuable in terms of contribution to clinical diagnosis of deafness in the Russian population and by highlighting the most common mutations, in addition to the GJB2 mutations, in this population. However, a panel of 33 genes includes only a very partial list of the known deafness genes and cannot be considered as “comprehensive genetic testing”. All studies today, and all NGS service laboratories are using panels of more than 100 deafness genes, and in the deafness variation database (https://deafnessvariationdatabase.org/) are listed 223 genes. Moreover, in order to perform “comprehensive genetic testing”, it is preferred to perform WES even if you intend to analyze just the deafness genes. Besides being cheaper, WES has the advantage of not having to update the panels frequently with new deafness genes, as all genes are included from the beginning.
Since the paper is valuable for the Russian population, and constitutes another proof that NGS methods, including panels (even non-comprehensive ones), are effective, I recommend publishing it with a different title. (For example: “Spectrum of genes for inherited hearing loss in the Russian population revealed by a targeted deafness gene panel").
The paper needs language editing.
Comments
General comments:
1) Submit all variants detected to ClinVar and indicate the ClinVar ID for each variant were requested.
2) Replace "genetic forms" with "genes", as "genetic forms" is a term usually used for modes of inheritance.
3) Be consistent with abbreviations.
Specific comments:
Abstract
Abbreviations should be used only if on the first time mentioned it is indicated what they stand for. Please do so for NGS, MPS, SHL.
L.23: “spectrum mutations” - should be “mutation spectrum”.
L.25: “mimic SHL genes" – if I understand correctly, it should be: "SHL genes that might be misdiagnosed as NSHL genes".
L.29: Remove << and >>.
L. 32, 33: Add the p. change for the SLC26A4 mutations. There should be consistency.
L.34: I suggest "cohort of patients" instead of "sample of patients".
Introduction
L. 69-70: "This method allows a lot of genes in one patient to be screened effectively" – I suggest: "This method allows effective screening of many genes in one test".
L.72: "The prevalence and spectrum" – I suggest to start the sentence with "Thus, the prevalence…".
L.76-83: “We enrolled 226 patients who were clinically diagnosed with sensorineural NSHL. In the three studies by Markova et al. and Lalayants et al. that present the results of the clinical description and audiological features of hearing loss caused by mutations in the STRC, USH2A, and OTOF genes, the patient sample was mostly formed from the same patients as in the present study [14,15]. The current study is intended to provide an understanding of the spectrum mutations in non-GJB2-related hearing loss in a cohort of Russian sensorineural NSHL patients and establish the best diagnostic algorithm.” – Paragraph not clear. I suggest to change it to: “We enrolled 226 patients who were clinically diagnosed with sensorineural NSHL. Part of this cohort was previously described in three studies conducted by Markova et al. and Lalayants et al. (what is the 3rd study?) that presented the phenotypes and the audiological features of hearing loss caused by mutations in the STRC, USH2A, and OTOF genes [14,15 the 3rd one missing]. The current study is intended to provide an understanding of the mutation spectrum in non-GJB2-related hearing loss in a cohort of Russian sensorineural NSHL patients and establish the best diagnostic algorithm.”
Please note that you indicated 3 studies but mentioned only 2. References 14 and 15 refer to STRC and OTOF. Please add the reference for USH2A.
Materials and Methods
L.87: “non-syndromic hearing loss” – should be “NSHL”. There should be consistency regarding all abbreviations.
L.87-88: “…with varying severity of hearing loss (1–4)” – I don’t understand what are the references (1-4) referring to. If these references do not refer to the cohort described in the sentence, they should be removed.
L.107-108: “[10] [16] [17] [18] [19]” – should be “[10, 16-19]”.
L.108: “Twenty-nine genes included in the panel” – 33 genes are listed.
L.121: “«NGSData»” – remove << and >>.
L.125: “incidence rate” – replace with “allele frequency”.
L.139-140: “Guidelines for interpretation of next-generation sequencing data [20,21] were used to define the clinical significance of newly discovered variants.” – I assume you mean the ACMG criteria analysis. ACMG guidelines should be applied according to Ricahrd et al., 2015 combined with the specifications of OZA et al., 2018. Please clarify that you mean the ACMG criteria based on these 2 references. In addition, please submit to ClinVar all reported variants and indicate submission reference for each variant in the Results section. Please add the ClinVar submission to Materials and Methods and to Results.
Results
L.145: ”… in 30 patients.” – Please add: “All variants were classified according to the ACMG criteria (2 references above) and submitted to ClinVar.”.
L.146: “Our MPS panel includes the STRC and OTOA genes, with the majority of mutant 146 alleles (50–77% percent in different ethnic groups) being gross deletions [22].“ – I suggest to change the sentence to: “Our MPS panel results revealed mutations in the STRC and OTOA genes, with the majority of mutant alleles (50–77% percent in different ethnic groups) being gross deletions [22].“.
L.148: “CNV” should be “CNVs”.
Figures 1,2: Not clear, too small, impossible to read.
L.174: “in the second stage” – did you mean “from the second stage”?
L.181: “the variants’ pathogenicity” – should be “: “the variants’ segregation”.
L.185-187: Needs language editing.
L.188-195: Add references for the previously described variants.
L.197-198: Reference missing for the “earlier study”.
L.202: “in a sample (4/226) of Russian patients” – should be “in X% (4/226) of our cohort of Russian patients”.
L.205-206: “approximately a quarter” – indicate X% instead.
L.216: “(ACMG)” – Please add: “(ACMG) and submitted to ClinVar”.
Table 2: Add a column: ClinVar ID
L.219: Instead of your reference 21, it should be OZA et al., 2018.
L.222: “Thirteen genetic forms” – What do you mean by “forms”? what kind of forms? Did you mean genes?
L.228-229: “GJB2 hearing loss” – remove quotation marks.
L.229-232: "was formed from the same cohort of patients, the proportion of the genetic forms studied in this work among all hereditary NSHL was 12%, significantly lower than that reported globally." – what is the size of the cohort? Did you mean 226+X? 12% - what is the number of patients? Did you mean the 48 solved patients mentioned above? Out of how many? If your panel was not performed for the total number, it would be wrong to count 48/226+X. Please clarify.
L.232-233: "the proportion of this form" – what do you mean by form? Did you mean the prevalence of non-GJB2 mutations? Please clarify.
L.238, 243: "forms" – do you mean other genes? Please clarify.
L.242: "using this" – replace with: "accordingly" or "bases on this".
L.242-243: " using this, we can calculate the total frequency of the studied non-GJB2-related forms among hereditary NSHL in the Russian Federation, i.e., 1 per 3500 newborns." – I suggest to replace with: "accordingly, in this study the prevalence of non-GJB2-related mutations would be 1 per 3500 newborns."
Fig.3: Instead of 1, 2, 3, please write the population origins.
L.248, 257: "genetic types" – replace with "genes".
L.248-254: Please start the paragraph with a sentence with the total number of genes detected in this study.
L.254: "was not different" – from what? Please clarify the sentence.
L.257-259: Sentence too complicated. I suggest: "The prevalence of the observed deafness genes above among the Russian population is similar to the prevalence reported in other populations".
L.259-260: Remove quotation marks.
Table 3:
L.262: "genetic types" – replace with "genes".
L.263: "among patients" – indicate the number of patients.
L.263-265: I don't understand the meaning of the *. Are those numbers taken from the reference toy indicated?
In the table: "Genetic Type" – replace with Genes.
The column "Among the 48 Cases of this Study (95%CI)" – I don't understand the (95%CI).
The column "In the World" – Are the number taken from the other study? As I wrote above, it's not clear.
L.273-275: "A familial analysis established the pathogenicity of five variants, allowing the cause of hearing loss to be determined in an additional six patients. Therefore, the overall diagnostic rate was 21% (48/226)." Are the additional six patients probands or are they relatives of solved probands? If they are relatives, they shouldn't be count. Please clarify.
L.282: "17 (23/48, 50%)" – Is it 17 or 23?
L.282-283: "an NSHL mimic" – replace with "a misdiagnosed SHL gene".
Discussion
L.292-294: Please add reference for the first sentence.
L.294-296: I don't understand how ACMG criteria is related to the other part of the sentence.
L.297: Replace "forms" with "genes".
L.299-300: "and mimic SHL 299 genes" - Should be: "and SHL genes misdiagnosed as SNHL genes".
L.301: "non-syndromic hearing loss" – Use abbreviation: NSHL. It should be consistent in the whole paper.
L.307: "21%" – add numbers (X/Y).
L.315: "among all hereditary non-syndromic hearing loss Russian patients was 12%," – Please indicate numbers and clarify the 12% according to my above comment for L.229-232.
L.316: "multi-ethnic sample of patients" – add the numbers of this cohort.
L.317, 324: Replace "genetic forms" with "genes".
L.323-324: "…and subsequent sequencing using an NGS panel, was higher than abroad at 55%. – the sentence is not clear. I suggest: "…and subsequent sequencing using an NGS panel, was 55%, higher than in the other studies mentioned above."
L.326: Replace "proportion" with "prevalence".
L.328: Add numbers for 30% (X/Y) and remove quotation marks.
L.330-332: "other TECTA MYO15A, POU3F4, TMPRSS3, PTPRQ, ADGRV1, TMC1, ACTG1, and OTOA genes accounted for less than 1%." – Change to: "other genes, TECTA MYO15A, POU3F4, TMPRSS3, PTPRQ, ADGRV1, TMC1, ACTG1, and OTOA, accounted for less than 1%."
L.334: Add "The" before "pathogenic".
L.335: Add "the" before "deletion".
L.336: "repeat" – did you mean "repeated"?
L.337: "were 25% and 15% mutations" – not clear.
L/352-355: "as s. Pendred, s. deafness, and male infertility (DIS) were also found." Should be: " as Pendred syndrome, deafness, and male infertility syndrome (DIS) were also found."
L.360: "GJB2" should be italics, "GJB2".
L.361: "connexin 26 gene" – should be: "GJB2 gene".
L.363-368: Add reference.
L.370: "which will identify two mutations in 51% of patients with NSHL " – you don't know that. This is the percentage identified, and based on it you predict detection of a large portion of patients.
P.372: " the time to diagnosis" – did you mean " the waiting time for diagnosis"?
L.376: "Of course, the optimal diagnostic algorithm varies among patients in different countries." – Today, WES is preferred for all populations.
Conclusions
L.380-381: "NSHL comprises a group of diseases that are difficult to diagnose the cause at the clinical level." – Sentence not clear.
L.381: Replace "forms" with "genes".
Sup. Table:
Correct reference 21 as mentioned above.
"sNSHL" – should be "SN NSHL" (SN – sensorineural).
The column: "HGMD Reference Number, References б " – should be replaced by ClinVar ID.
References
Correct ref. 21.
Author Response
Dear reviewer! Thank you very much for the thorough analysis of our article. I think all the corrections will benefit the article! All changes are highlighted in blue. The corrections requested by another reviewer are highlighted in yellow. Corrections to similar comments are highlighted in green.
We changed the title of the article to "Spectrum of genes for Non-GJB2-Related Non-Syndromic Hearing Loss in the Russian population revealed by a targeted deafness gene panel"
General comments:
- Submit all variants detected to ClinVar and indicate the ClinVar ID for each variant were requested.
Answer: We uploaded all variants to ClinVar. For the novel variants we indicated The ClinVar accession numbers but not ClinVar ID, because our records will be public later (as the СlinVar support said) and there are no ClinVar ID for them at this moment.
2) Replace "genetic forms" with "genes", as "genetic forms" is a term usually used for modes of inheritance.
Answer: We replaced "genetic forms" with "genes".
3) Be consistent with abbreviations.
Answer: We fixed it
Specific comments:
Abstract
Abbreviations should be used only if on the first time mentioned it is indicated what they stand for. Please do so for NGS, MPS, SHL.
Answer: We replaced the abbreviation NGS with MPS and indicated what MPS and SHL stand for in the entire article.
L.23: “spectrum mutations” - should be “mutation spectrum”.
Answer: We corrected it (Line 25).
L.25: “mimic SHL genes" – if I understand correctly, it should be: "SHL genes that might be misdiagnosed as NSHL genes".
Answer: Yes, that's right. We added your explanation to the abstract (Lines 27-28).
L.29: Remove << and >>.
Answer: We corrected it (Line 32).
- 32, 33: Add the p. change for the SLC26A4mutations. There should be consistency.
Answer: We added it (line 36).
L.34: I suggest "cohort of patients" instead of "sample of patients".
Answer: We corrected it (Line 38).
Introduction
- 69-70: "This method allows a lot of genes in one patient to be screened effectively" – I suggest: "This method allows effective screening of many genes in one test".
Answer: We corrected it (Lines 71-72).
L.72: "The prevalence and spectrum" – I suggest to start the sentence with "Thus, the prevalence…".
Answer: We corrected it (Line 74).
L.76-83: “We enrolled 226 patients who were clinically diagnosed with sensorineural NSHL. In the three studies by Markova et al. and Lalayants et al. that present the results of the clinical description and audiological features of hearing loss caused by mutations in the STRC, USH2A, and OTOF genes, the patient sample was mostly formed from the same patients as in the present study [14,15]. The current study is intended to provide an understanding of the spectrum mutations in non-GJB2-related hearing loss in a cohort of Russian sensorineural NSHL patients and establish the best diagnostic algorithm.” – Paragraph not clear. I suggest to change it to: “We enrolled 226 patients who were clinically diagnosed with sensorineural NSHL. Part of this cohort was previously described in three studies conducted by Markova et al. and Lalayants et al. (what is the 3rd study?) that presented the phenotypes and the audiological features of hearing loss caused by mutations in the STRC, USH2A, and OTOF genes [14,15 the 3rd one missing]. The current study is intended to provide an understanding of the mutation spectrum in non-GJB2-related hearing loss in a cohort of Russian sensorineural NSHL patients and establish the best diagnostic algorithm.”
Please note that you indicated 3 studies but mentioned only 2. References 14 and 15 refer to STRC and OTOF. Please add the reference for USH2A.
Answer: We corrected it and added a link to the third article (Lines 79-83).
Materials and Methods
L.87: “non-syndromic hearing loss” – should be “NSHL”. There should be consistency regarding all abbreviations.
Answer: We corrected this throughout the article.
L.87-88: “…with varying severity of hearing loss (1–4)” – I don’t understand what are the references (1-4) referring to. If these references do not refer to the cohort described in the sentence, they should be removed.
Answer: We removed it (Line 89).
L.107-108: “[10] [16] [17] [18] [19]” – should be “[10, 16-19]”.
Answer: We corrected it (Line 110).
L.108: “Twenty-nine genes included in the panel” – 33 genes are listed.
Answer: 29 genes are genes responsible for the development of non-syndromic forms of hearing loss. The other four genes are only responsible for the development of syndromic forms of hearing loss. We have corrected the following sentence to make it clearer (Lines 111-113).
L.121: “«NGSData»” – remove << and >>.
Answer: We removed it (Line 123).
L.125: “incidence rate” – replace with “allele frequency”.
Answer: We corrected it (Line 127).
L.139-140: “Guidelines for interpretation of next-generation sequencing data [20,21] were used to define the clinical significance of newly discovered variants.” – I assume you mean the ACMG criteria analysis. ACMG guidelines should be applied according to Ricahrd et al., 2015 combined with the specifications of OZA et al., 2018. Please clarify that you mean the ACMG criteria based on these 2 references. In addition, please submit to ClinVar all reported variants and indicate submission reference for each variant in the Results section. Please add the ClinVar submission to Materials and Methods and to Results.
Answer: We clarified that we mean the ACMG criteria based on these 2 references. Also we added ClinVar submissions.
Results
L.145: ”… in 30 patients.” – Please add: “All variants were classified according to the ACMG criteria (2 references above) and submitted to ClinVar.”.
Answer: We did it (Line 151-152).
L.146: “Our MPS panel includes the STRC and OTOA genes, with the majority of mutant 146 alleles (50–77% percent in different ethnic groups) being gross deletions [22].“ – I suggest to change the sentence to: “Our MPS panel results revealed mutations in the STRC and OTOA genes, with the majority of mutant alleles (50–77% percent in different ethnic groups) being gross deletions [22].“.
Answer: Here we are not talking about our results, but about the world (explain why we decided to hold this stage). We corrected this sentence and hope it will be clearer: «Our MPS panel includes the STRC and OTOA genes. According to global data, the majority of mutant alleles (50–77% percent in different ethnic groups) being gross deletions» (Lines 153-155).
L.148: “CNV” should be “CNVs”.
Answer: We have corrected this throughout the text.
Figures 1,2: Not clear, too small, impossible to read.
Answer: We increased the size and resolution.
L.174: “in the second stage” – did you mean “from the second stage”?
Answer: Yes. We corrected it (Line 181).
L.181: “the variants’ pathogenicity” – should be “: “the variants’ segregation”.
Answer: We corrected it (Line 189).
L.185-187: Needs language editing.
Answer: We corrected it (Lines 195-197): «Thus, after all steps, among 226 patients with non-GJB2 SNHL, a genetic cause of the disease was found in 48 (21%). A total of 55 pathogenic/likely pathogenic variants were identified (see Supplementary Table S1)».
L.188-195: Add references for the previously described variants
Answer: We added references (Lines 202-203).
L.197-198: Reference missing for the “earlier study”.
Answer: The link is at the end of the sentence (Line 210).
L.202: “in a sample (4/226) of Russian patients” – should be “in X% (4/226) of our cohort of Russian patients”.
Answer: We corrected it (Line 212).
L.205-206: “approximately a quarter” – indicate X% instead.
Answer: We added X% (Line 215).
L.216: “(ACMG)” – Please add: “(ACMG) and submitted to ClinVar”.
Answer: We did it (Line 225).
Table 2: Add a column: ClinVar ID
Answer: Because our records will be public later (as the СlinVar support said) and there are now ClinVar ID for the novel variants at this moment, we added The ClinVar accession numbers of variants in the Table 2.
L.219: Instead of your reference 21, it should be OZA et al., 2018.
Answer: We changed it (Line 228).
L.222: “Thirteen genetic forms” – What do you mean by “forms”? what kind of forms? Did you mean genes?
Answer: We replaced "genetic forms" with "genes" (Line 231).
L.228-229: “GJB2 hearing loss” – remove quotation marks.
Answer: We removed it (Line 294).
Another reviewer asked that this paragraph be moved to the Discussion section so there would be no repetition. Therefore, this and the next six comments have been corrected in the Discussion section.
L.229-232: "was formed from the same cohort of patients, the proportion of the genetic forms studied in this work among all hereditary NSHL was 12%, significantly lower than that reported globally." – what is the size of the cohort? Did you mean 226+X? 12% - what is the number of patients? Did you mean the 48 solved patients mentioned above? Out of how many? If your panel was not performed for the total number, it would be wrong to count 48/226+X. Please clarify.
Answer: We have rewritten this paragraph (Line 292-300): «The total contribution of the genes studied in this work to non-GJB2-related hearing loss was 21% (48/226). Previously, we investigated the mutation spectrum and the frequency of GJB2 hearing loss among Russian patients with NSHL [34]. According to Bliznetz et al., the frequency of GJB2 mutations is 43%. Thus, the frequency of non-GJB2-related hearing loss is 57%, according to this study. Our cohort of Russian patients investigated using the MPS panel is part of the one described in this article. Accordingly, the estimated proportion of the genes studied in this work among all hereditary NSHL by taking 21% of the 57%. The total contribution the studied genes all hereditary NSHL is equal to 12%, significantly lower than that reported globally». We indicated that the 12% frequency is estimated because we calculated it. We think we can use it because our patient cohort was randomly selected from the cohort described in Bliznetz et al.
L.232-233: "the proportion of this form" – what do you mean by form? Did you mean the prevalence of non-GJB2 mutations? Please clarify.
Answer: We have rewritten this paragraph (Lines 300-303): «In a multi-ethnic sample of 1119 patients from Europe, America, and Asia, the proportion of the genes studied in this work was equal to 29% (ꭓ2 = 21.191, p < 0.001) [10], and, in a sample of 207 patients from France, it was 23% (ꭓ2 = 12.120, p < 0.001)».
L.238, 243: "forms" – do you mean other genes? Please clarify.
Answer: Yes, we meant genes. Unfortunately, we deleted this sentence because a similar sentence was already in the discussion section.
L.242: "using this" – replace with: "accordingly" or "bases on this".
Answer: We replaced "using this" with "accordingly" in the Discussion section (Line 313).
L.242-243: " using this, we can calculate the total frequency of the studied non-GJB2-related forms among hereditary NSHL in the Russian Federation, i.e., 1 per 3500 newborns." – I suggest to replace with: "accordingly, in this study the prevalence of non-GJB2-related mutations would be 1 per 3500 newborns."
Answer: We replaced it in the Discussion section (Line 313-314).
Fig.3: Instead of 1, 2, 3, please write the population origins.
Answer: We did it and transfer Fig.3 in the Discussion section.
L.248, 257: "genetic types" – replace with "genes".
Answer: We replaced it.
L.248-254: Please start the paragraph with a sentence with the total number of genes detected in this study.
Answer: We added the sentence: «The total number of genes detected in this study is thirteen» (Line 236).
L.254: "was not different" – from what? Please clarify the sentence.
Answer: We deleted this sentence.
L.257-259: Sentence too complicated. I suggest: "The prevalence of the observed deafness genes above among the Russian population is similar to the prevalence reported in other populations".
Answer: We replaced it (Lines 244-246).
L.259-260: Remove quotation marks.
Answer: We did it (Line 247).
Table 3:
L.262: "genetic types" – replace with "genes".
Answer: We did it (Line 249).
L.263: "among patients" – indicate the number of patients.
Answer: We added this information (Line 250).
L.263-265: I don't understand the meaning of the *. Are those numbers taken from the reference toy indicated?
Answer: Yes, the comparison was made with the cohort of patients from the indicated article (Sloan-Heggen, C.M. et al.). A little paraphrased: «The comparison was made between the proportions in our cohort of Russian patients and the multi-ethnic sample of patients from Europe, America, and Asia from the study conducted by Sloan-Heggen et al.». We hope that makes more sense (Lines 250-252).
In the table: "Genetic Type" – replace with Genes.
Answer:We did it.
The column "Among the 48 Cases of this Study (95%CI)" – I don't understand the (95%CI).
Answer: 95%CI means a сonfidence interval with the 95% confidence level. We indicated what "CI" means in the table.
The column "In the World" – Are the number taken from the other study? As I wrote above, it's not clear.
Answer: We answered above. We rewrite this column.
L.273-275: "A familial analysis established the pathogenicity of five variants, allowing the cause of hearing loss to be determined in an additional six patients. Therefore, the overall diagnostic rate was 21% (48/226)." Are the additional six patients probands or are they relatives of solved probands? If they are relatives, they shouldn't be count. Please clarify.
Answer: These probands are not related to the others. They are included in the original cohort, but prior to the family analysis, the pathogenicity of their variants was not proven and, therefore, the diagnosis was not made. Family analysis was performed to clarify the pathogenicity of some VUS variants (e.g., determination of trans position). We write about this at the end of paragraph 3.1 in the results section (Lines 189-193): «A family analysis was performed to analyze segregation of the variants in parents and other relatives when variants of uncertain significance were found. It helped us to use the ACMG criteria to classify these variants. Some families were not analyzed due to the lack of biological material of relatives. Trans-position for six variants was shown by examining the probands’ parents in seven families».
L.282: "17 (23/48, 50%)" – Is it 17 or 23?
Answer: 23. We corrected it and moved to the discussion section because another reviewer asked (Line 341).
L.282-283: "an NSHL mimic" – replace with "a misdiagnosed SHL gene".
Answer: 23. We did it and moved to the discussion section (Line 342).
Discussion
L.292-294: Please add reference for the first sentence.
Answer: We added it (Line 256).
L.294-296: I don't understand how ACMG criteria is related to the other part of the sentence.
Answer: We removed the ACMG part of the sentence (Lines 258-259).
L.297: Replace "forms" with "genes".
Answer: We did it (Line 260).
L.299-300: "and mimic SHL 299 genes" - Should be: "and SHL genes misdiagnosed as SNHL genes" (Lines 260-261).
Answer: We did it (Lines 262-263).
L.301: "non-syndromic hearing loss" – Use abbreviation: NSHL. It should be consistent in the whole paper.
Answer: We corrected this throughout the manuscript.
We rewrote this paragraph because the reviewer asked us to merge it with the data from section 3.4 of the results to avoid repetition.
L.307: "21%" – add numbers (X/Y).
Answer: We added it (Line 274).
L.315: "among all hereditary non-syndromic hearing loss Russian patients was 12%," – Please indicate numbers and clarify the 12% according to my above comment for L.229-232.
Answer: We responded to your previous comment and rephrased the sentence (Lines 295-297): «According to Bliznetz et al., the estimated contribution of the studied genes among all hereditary NSHL Russian patients was 12%...».
L.316: "multi-ethnic sample of patients" – add the numbers of this cohort.
Answer: We added it (Lines 300-303): « In a multi-ethnic sample of 1119 patients from Europe…».
L.317, 324: Replace "genetic forms" with "genes".
Answer: We did it.
L.323-324: "…and subsequent sequencing using an NGS panel, was higher than abroad at 55%. – the sentence is not clear. I suggest: "…and subsequent sequencing using an NGS panel, was 55%, higher than in the other studies mentioned above."
Answer: Because we had to merge this part with the part from the results section, we rewrote this sentence (Lines 308-311): «and subsequent sequencing using an MPS panel, was 55%, whereas in a multi-ethnic sample of 1119 patients from Europe, America, and Asia, this was 37%...».
L.326: Replace "proportion" with "prevalence".
Answer: We did it (Line 319).
L.328: Add numbers for 30% (X/Y) and remove quotation marks.
Answer: We did it (Line 321).
L.330-332: "other TECTA MYO15A, POU3F4, TMPRSS3, PTPRQ, ADGRV1, TMC1, ACTG1, and OTOA genes accounted for less than 1%." – Change to: "other genes, TECTA MYO15A, POU3F4, TMPRSS3, PTPRQ, ADGRV1, TMC1, ACTG1, and OTOA, accounted for less than 1%."
Answer: We did it (Line 324).
L.334: Add "The" before "pathogenic".
Answer: We did it (Line 327).
L.335: Add "the" before "deletion".
Answer: We did it (Line 328).
L.336: "repeat" – did you mean "repeated"?
Answer: Yes. We corrected it (Line 330).
L.337: "were 25% and 15% mutations" – not clear.
Answer: Replaced the sentence with (Lines 330-331): «Two repeated mutations c.107A>C and c.1001G>T, were 25% and 15% of mutant alleles of the SLC26A4 gene, respectively».
L/352-355: "as s. Pendred, s. deafness, and male infertility (DIS) were also found." Should be: " as Pendred syndrome, deafness, and male infertility syndrome (DIS) were also found."
Answer: We corrected it (Lines 347-349).
L.360: "GJB2" should be italics, "GJB2".
Answer: We did it (Line 358).
L.361: "connexin 26 gene" – should be: "GJB2 gene".
Answer: We did it (Line 359).
L.363-368: Add reference.
Answer: We added a reference to the previous sentence. This sentence is already about our research. We indicated that (Line 361).
L.370: "which will identify two mutations in 51% of patients with NSHL " – you don't know that. This is the percentage identified, and based on it you predict detection of a large portion of patients.
Answer: Replaced the sentence with (Lines 367-368): «… which is probably to detect two mutations in a significant proportion of patients with NSHL using MLPA methods».
P.372: " the time to diagnosis" – did you mean " the waiting time for diagnosis"?
Answer: Yes. We corrected it (Lines 369-370).
L.376: "Of course, the optimal diagnostic algorithm varies among patients in different countries." – Today, WES is preferred for all populations.
Answer: We rewrote that part (Lines 373-375): «Of course, whole exome sequencing is currently the preferred method for diagnosing hereditary hearing loss for all population because it allows identify variants in all known hearing loss genes. However, the country-specific algorithms can reduce the cost of diagnosis and speed up diagnosis in many cases».
Conclusions
L.380-381: "NSHL comprises a group of diseases that are difficult to diagnose the cause at the clinical level." – Sentence not clear.
Answer: Replaced the sentence with (Lines 378-379): «NSHL is a heterogeneous group of diseases, the etiological diagnosis of which is difficult to establish by traditional clinical examination».
L.381: Replace "forms" with "genes".
Answer: We corrected it (Line 380).
Sup. Table:
Correct reference 21 as mentioned above.
Answer: We corrected it.
"sNSHL" – should be "SN NSHL" (SN – sensorineural).
Answer: We corrected it.
The column: "HGMD Reference Number, References б " – should be replaced by ClinVar ID.
Answer: We added ClinVar ID for the described earlier in the ClinVar variants. For the novel variants , we added The ClinVar accession numbers.
References
Correct ref. 21.
Answer: We corrected it.

Round 2
Reviewer 1 Report
The authors have addressed most of my comments. This revised version is almost ready for publication. I have only two additional suggestions:
1) Lines 36-38. The sentence is still not very clear, and 'Syndromic genes' is not an appropriate term. I suggest to rephrase it as follows:
"Pathogenic variants in genes involved in SHL accounted for almost half of the cases with an established molecular genetic diagnosis, which was 10% of..."
2) A typo in line 83. 'understanding of the spectrum mutation spectrum' should read 'understanding of the mutation spectrum'.
Author Response
Dear Reviewer. We have corrected the remaining remarks. Thank you very much for thoroughly reviewing the article!
1) Lines 36-38. The sentence is still not very clear, and 'Syndromic genes' is not an appropriate term. I suggest to rephrase it as follows:
"Pathogenic variants in genes involved in SHL accounted for almost half of the cases with an established molecular genetic diagnosis, which was 10% of..."
Answer: We replaced this sentence (Lines 36-38). It makes so much more sense. Thank you for the recommendation!
2) A typo in line 83. 'understanding of the spectrum mutation spectrum' should read 'understanding of the mutation spectrum'.
Answer: We fixed it (Line 83).

Reviewer 2 Report
1) I did not see a cover letter referring to all corrections, which made it harder, and it took more time, to re-review the paper.
2) The paper still needs professional language editing. Some of the sentences are too complicated and hard to understand.
3) I listed several comments below, however, major parts, particularly the discussion, have to be revised, with special attention to the prevalence in the compared cohorts, as the numbers and percentages are not clear.
Abstract
L.28: “(mimic SHL)” – mimic SHL means that the genes mimic SHL, while what you mean is that these are SHL genes mimicking NSHL (look like NSHL). The term as used is not correct. Remove it please.
Introduction
L.82: remove “spectrum” before “mutation spectrum”.
Results
L. 150-151: remove “(2 references above)”.
L.189-191: “A family analysis was performed to analyze the variants’ segregation in parents and other relatives when variants of uncertain significance were found. It helped us to use the ACMG criteria to classify these variants.” – I suggest: “For variants of uncertain significance, a family analysis was performed to confirm the variants’ segregation with the HL among the family members”.
L.196: “in 48 (21%)” – please add “cases” or “patients”.
Table 3
Second column: “Genes of HL” – should be “Phenotype”.
“The Proportion of Probands with an Established Genes of Hearing Loss” – remove “an”.
Discussion
Should be re-written. Parts of it belong or repeat the results, including Figure 3). Many of the sections are not clear.
L.277: “family cases” – should be “familial cases”.
L.295: “According to Bliznetz et al.,” – I suggest: “According to this study of Bliznetz et al.,”
L.295-296: “Thus, the frequency of non-GJB2-related hearing loss is 57%, according to this study.” – what do you mean? If I understood correctly, the previous sentence and this one should be combined to: “According to this study of Bliznetz et al., the frequency of GJB2 mutations is 43%, and frequency of non-GJB2-related hearing loss is 57%.”. Is it correct?
L.296-302: “Our cohort of Russian patients investigated using the MPS panel is part of the one described in this article (What part? Should add numbers: 226 out of X of the other study). Accordingly, the estimated proportion of the genes studied in this work among all hereditary NSHL by taking 21% of the 57% (I don’t understand what you mean). The total contribution the studied genes all hereditary NSHL is equal to 12% (not clear), significantly lower than that reported globally. In a multi-ethnic sample of 1119 patients from Europe, America, and Asia, the proportion of the genes studied in this work - Please remove these words was equal to 29%” – this paragraph is not clear at all.
L.302-314: the yellow section needs editing.
Figure 3 should be part of the results.
Figure 3 legend: Remove 1-3. No need to repeat the information. Add the references [10] and [11] under the bars.
Author Response
Dear reviewer! All changes in the manuscript are highlighted in blue. The corrections requested by another reviewer are highlighted in yellow. Corrections to similar comments are highlighted in green.
At the previous round of the review we sent a detailed response to your comments with the line numbers for each correction. You can find our answer in section "Authors' Responses to Reviewer's Comments" of the first round of the review. Answers to each your comment were after the word "Answer:".
We tried to take all your comments and rewrote the paragraph about the prevalence in the compared cohorts. We would like to note that this manuscript has undergone English language editing by MDPI, before it was submitted to this journal.
Below we responded to each of your comments.
Abstract
L.28: “(mimic SHL)” – mimic SHL means that the genes mimic SHL, while what you mean is that these are SHL genes mimicking NSHL (look like NSHL). The term as used is not correct. Remove it please.
Answer: We removed it (Line 28).
Introduction
L.82: remove “spectrum” before “mutation spectrum”.
Answer: We removed it (Line 83).
Results
- 150-151: remove “(2 references above)”.
Answer: We removed it (Lines 150-151).
L.189-191: “A family analysis was performed to analyze the variants’ segregation in parents and other relatives when variants of uncertain significance were found. It helped us to use the ACMG criteria to classify these variants.” – I suggest: “For variants of uncertain significance, a family analysis was performed to confirm the variants’ segregation with the HL among the family members”.
Answer: We replaced this sentence (Lines 189-190).
L.196: “in 48 (21%)” – please add “cases” or “patients”.
Answer: We did it (Line 195).
Table 3
Second column: “Genes of HL” – should be “Phenotype”.
Answer: We replaced it.
“The Proportion of Probands with an Established Genes of Hearing Loss” – remove “an”.
Answer: We removed «an».
Discussion
Should be re-written. Parts of it belong or repeat the results, including Figure 3). Many of the sections are not clear.
Answer: It seems to us, as well as to another reviewer, that it is more logical to leave these calculations in the discussion, in the same place where comparisons with the data of other authors are made. In the results we have briefly written our data.
L.277: “family cases” – should be “familial cases”.
Answer: We replaced it (Line 280).
L.295: “According to Bliznetz et al.,” – I suggest: “According to this study of Bliznetz et al.,”
Answer: We did it (Line 298).
L.295-296: “Thus, the frequency of non-GJB2-related hearing loss is 57%, according to this study.” – what do you mean? If I understood correctly, the previous sentence and this one should be combined to: “According to this study of Bliznetz et al., the frequency of GJB2 mutations is 43%, and frequency of non-GJB2-related hearing loss is 57%.”. Is it correct?
Answer: It is correct. We combined these sentences as you suggested (Lines 298-300): «the frequency of GJB2 muta299 tions is 43% (1104/2569), and the frequency of non-GJB2-related hearing loss is 57% 300 (1465/2569)».
L.296-302: “Our cohort of Russian patients investigated using the MPS panel is part of the one described in this article (What part? Should add numbers: 226 out of X of the other study). Accordingly, the estimated proportion of the genes studied in this work among all hereditary NSHL by taking 21% of the 57% (I don’t understand what you mean). The total contribution the studied genes all hereditary NSHL is equal to 12% (not clear), significantly lower than that reported globally. In a multi-ethnic sample of 1119 patients from Europe, America, and Asia, the proportion of the genes studied in this work - Please remove these words was equal to 29%” – this paragraph is not clear at all.
Answer: We have completely rewritten this paragraph. We kept the 29%, but wrote it more clearly with numbers of cases. All of these results are also shown in Figure 3. We were asked to move the paragraph to the Discussion section, so it's here.
L.302-314: the yellow section needs editing.
Answer: We have corrected this entire paragraph.
Figure 3 should be part of the results.
Answer: We did it.
Figure 3 legend: Remove 1-3. No need to repeat the information. Add the references [10] and [11] under the bars.
